# Identifying the Microbiome of the Adenoid Surface of Children Suffering from Otitis Media with Effusion and Children without Middle Ear Effusion Using 16S rRNA Genetic Sequencing

**DOI:** 10.3390/microorganisms11081955

**Published:** 2023-07-31

**Authors:** Oļegs Sokolovs-Karijs, Monta Brīvība, Rihards Saksis, Maija Rozenberga, Francesca Girotto, Jana Osīte, Aigars Reinis, Gunta Sumeraga, Angelika Krūmiņa

**Affiliations:** 1Department of Otolaryngology, Riga Stradiņš University, 16 Dzirciema Str., LV-1007 Riga, Latvia; 2AIWA Clinic, 241 Maskavas Str., LV-1019 Riga, Latvia; 3Latvian Biomedicine Research and Study Center, 1 Ratsupites Str., LV-1067 Riga, Latvia; 4Faculty of Medicine, Riga Stradiņš University, 16 Dzirciema Str., LV-1007 Riga, Latvia; 5Centrālā Laboratorrija, 1b. Šarlotes Str., LV-1011 Riga, Latvia; 6Department of Biology and Microbiology, Riga Stradiņš University, 16 Dzirciema Str., LV-1007 Riga, Latvia; 7Department of Infectology, Riga Stradiņš University, 16 Dzirciema Str., LV-1007 Riga, Latvia

**Keywords:** bacteria, adenoids, otitis media with effusion, microbiome, 16S rRNA genetic sequencing

## Abstract

Background: The upper respiratory tract harbors diverse communities of commensal, symbiotic, and pathogenic organisms, originating from both the oral and nasopharyngeal microbiota. Among the primary sites of microbial colonization in the upper airways are the adenoids. Alterations in the adenoid microbiota have been implicated in the development of various conditions, including secretory otitis media. Aim: This study aims to employ 16S rRNA genetic sequencing to identify the most common bacteria present on the surface of adenoids in children with otitis media with effusion and compare them with children without pathologies in the tympanic cavity. Additionally, we seek to determine and compare the bacterial diversity in these two study groups. Materials and Methods: A total of nineteen samples from the adenoid surfaces were collected, comprising two groups: thirteen samples from children without middle ear effusion and six samples from children with secretory otitis media. The libraries of the V3–V4 hypervariable region of the bacterial 16S rRNA gene was made and sequenced using MiSeq platform. Results: The most prevalent phyla observed in both groups were Proteobacteria, Firmicutes, and Bacteroidetes. The most common bacterial genera identified in both groups were *Haemophilus*, *Streptococcus*, *Moraxella*, *Fusobacterium*, and *Bordetella*, with *Fusobacterium* and Moraxella being more prevalent in the groups that had no middle ear effusion, while *Haemophulus* and *Streptococcus* were more prevalent in the otitis media with effusion group, although not in a statistically significant way. Statistical analysis shows a trend towards bacterial composition and beta diversity being similar between the study groups; however, due to the limited sample size and unevenness between groups, we should approach this data with caution. Conclusion: The lack of prolific difference in bacterial composition between the study groups suggests that the role of the adenoid microbiome in the development of otitis media with effusion may be less significant.

## 1. Introduction

The upper respiratory tract is lined by respiratory epithelium; constantly exposed to microorganisms, it serves as a crucial defense mechanism against environmental pathogens and irritants [1].

Moreover, the upper respiratory tract harbors different communities of commensal, symbiotic, and pathogenic organisms, both from oral and nasopharyngeal microbiota. The microbiota significantly contributes to the proper development, training, and function of the host’s immune system. It is involved in the regulation of immune responses and can help to educate the immune system to recognize and respond appropriately to various threats, such as harmful pathogens [2].

Factors like the use of antibiotics, age, lifestyle, and changes in diet primarily influence the composition and function of human microbiota. Changes could upset the microbial interactions and, consequently, the maintenance of physiological balance, which could lead to a weakened immune system [3,4].

The most common types of bacteria present in healthy individuals include Firmicutes, Bacteroidetes, Actinobacteria, and Proteobacteria. These phyla are dominated by specific genera such as *Bifidobacterium*, *Corynebacterium*, *Staphylococcus*, *Streptococcus*, *Haemophilus*, *Dolosigranulum*, and *Moraxella.* In the oropharynx, bacterial communities are more diverse and dominated by *Streptococcal species*, *Neisseria* spp., *Rothia* spp., and anaerobic bacteria [5,6].

Changes in the oral or nasopharyngeal microbiota can contribute to the development of various conditions; these alterations have been associated with an increased risk of several pathologies such as otitis media and secretory otitis media [7]. Otitis media with effusion is caused by fluid accumulation in the middle ear behind the tympanic membrane, without a clear clinical presentation suggesting acute inflammation. This pathology is prevalent in childhood globally, typically affecting around 80% of children by the age of ten. Recurrent infections may contribute to causing tissue hypertrophy and Eustachian tube dysfunction [8,9].

Among the primary sites of microbial accumulation in the body are adenoids, or pharyngeal tonsils, which contain specialized immune cells, such as lymphocytes, macrophages, and dendritic cells, making them essential organs responsible for both mucosal and systemic adaptive immunity of the upper airways. 

The adenoids, due to their constant exposure to allergens and pathogens, and because of their immunological function and location, serve as reservoirs for viruses and bacteria. Therefore, adenoid’s microbiota refers to a complex community of microorganisms that inhabit this lymphoid tissue that can influence the pathogenesis of respiratory infections [8].

A condition frequently observed in children affected by otitis media with effusion is adenoid hypertrophy. The enlargement of adenoid tissue can block the nasopharyngeal orifice of the Eustachian tube, leading to negative pressure within the middle ear cavity and mucosal changes. The most common bacterial pathogen found is *H. influenzae*, followed by *Streptococcus pneumoniae* and *Moraxella catarrhalis* [9,10].

The attachment of microbial cells to a surface, followed by their colonization and maturation, results in the formation of biofilms, which are also related to many pathological conditions involving the upper respiratory tract. 

These surface-associated microbial communities are enclosed in a self-produced extracellular polymeric substance matrix, mostly consisting of polysaccharides, proteins, and extracellular-DNA that provide a surface for bacterial adhesion.

A biofilm matrix is able to protect the underlying microorganisms from pH alterations, osmolarity, nutrient undersupply, and mechanical forces, as well as blocking the access from antibiotics and the host’s immune cells [1,11].

As a result of this additional resistance mechanism, the formation of bacterial biofilms has shown a higher resistance to antibiotics and, therefore, it has been associated with the development of chronic and recurrent microbial infections (recorded in at least 60% of cases). 

The bacterial species found most often involved in biofilm-associated respiratory tract infections are *Klebsiella pneumonia*, *Pseudomonas aeruginosa*, *Staphylococcus* spp., *Escherichia coli*, and Haemophilus influenza, in addition to a few *Mycobacteria* spp., like *M. tuberculosis* [11,12].

In parallel, recent advances in molecular techniques have emerged, revolutionizing clinical microbiology studies by allowing the identification and description of microbiomes with higher accuracy compared with the traditional culture-based methods. 

The microbiome of adenoids can be analyzed through a Polymerase Chain Reaction that enhances pathogen detection, as well as 16S ribosomal RNA gene sequencing that can identify and determine the composition of the different bacterial species present in the tissues.

Furthermore, this method facilitates the comparison of the phylogeny and taxonomy of various pathogens in different regions, thereby aiding in the identification of potential connections with different pathologies and outcomes [9,13].

## 2. Materials and Methods

### 2.1. Ethics Statement

This research is compliant with the Helsinki convention and conducted with the approval of Riga Stradiņš university ethics committee (approval nr. 2-PĒĶ-4/264/2022). All participants or their official representatives signed an agreement to participate in this scientific project.

### 2.2. Patient Selection, Inclusion, and Exclusion Criteria

The research material for this study was obtained from adenoid tissue collected during planned adenotomy procedures in children aged 3 to 7 years. The adenotomy surgeries were performed by certified surgeons at the multifunctional outpatient clinic “AIWA clinic” under general anesthesia, following the indications for adenoid tissue removal. All sample collection took place between October and November 2022. The samples were categorized into two groups: patients with adenoid hyperplasia but no fluid in the middle ear, and patients with adenoid hyperplasia and fluid in the middle ear. We should emphasize that indications for adenoid surgeries performed on children with no middle ear effusion include the potential prevention of said middle ear effusion forming by improving ventilation though the eustachian tube and reducing bacterial load in the nasopharynx.

#### 2.2.1. Inclusion Criteria for Samples Collected from Patients without Middle Ear Effusion

Age of patients: 3 to 7 years old;A-type tympanometry bilaterally (no evidence of middle ear fluid accumulation);Indications for adenoid surgery based on history of frequent recurrent upper airway infections, obstructive sleep apnea episodes, and maxillofacial anomalies;Endoscopical findings—adenoid tissue hyperplasia 2/3/4 grade;Operation under general anesthesia;The patient and his/her representative agree to participate in the study.

#### 2.2.2. Inclusion Criteria for Samples Collected from Patients with Middle Ear Effusion

Age of patients: 3 to 7 years old;B-type tympanometry unilaterally or bilaterally (evidence of middle ear effusion);Indications for adenoid surgery based on history of frequent recurrent upper airway infections, obstructive sleep apnea episodes, maxillofacial anomalies, frequent middle ear infections, and prolonged hearing impairment due to conductive hearing loss;Endoscopic findings—adenoid tissue hyperplasia 2/3/4 grade;Operation under general anesthesia;The patient and his/her representative agree to participate in the study.

#### 2.2.3. Exclusion Criteria for Both Groups

Patients over the age of 7 and under the age of 3;Patients from other geographical locations traveling to Latvia specifically for the operation;Immunocompromised patients: HIV positive patients, patients with hepatitis A/B/C, patients undergoing chemotherapy due to oncological diseases, *diabetes mellitus* patients, patients with chronic autoimmune diseases (sarcoidosis, Wegner’s granulomatosis);Patients with cleft malformations;Patients who received systemic antibiotic treatment for the last 2 weeks before the operation;Patients with signs of acute upper airway infections: fever >38 °C, purulent discharge from the nose and/or ear canals, productive cough, overall fatigue and shimmers;Patients who receive immunomudolators—microflora altering substances, for example, bifido-bacterium food supplements.

### 2.3. Material Collection 

Samples were collected using microbiological swabs from the surface of the adenoid tissue removed during an adenotomy surgical procedure. The surgery commenced under general anesthesia. Prior to adenoid removal, the surgeon disinfected the surface of the lips and skin around the lips using the surface disinfectant available at that time; after that, the surgeon carefully disinfected the oral cavity using chlorohexidine-based disinfection solution for mucosal surfaces. This internal disinfection is not excessive, so as to not allow the chlorohexidine fluid to irrigate the adenoid surface and impact the natural macroflora. Foregoing the oral mucosal disinfection is required to reduce the risk of contamination of dental and buccal microorganisms while evacuating the adenoid tissue from the nasopharynx through the oral cavity and outside—through the mouth. The adenoid tissue was removed using adenotomy curette under nasal endoscopic control. After the removal, the adenoid tissue was placed on a sterile surgical table. A cotton swab was placed on the adenoid surface and rotated multiple times. The tip of the cotton swab was placed in the “COPAN-Enat” system and immediately transported to the Biomediacal research and study center for preservation and sequencing. 

### 2.4. DNA Extraction and Sequencing

The FastDNA Spin Kit for Soil (MP Biomedicals, Santa Ana, CA, USA) was used for the microbial DNA extraction of the frozen adenoid swabs. For the metagenomic analysis sequencing of the V3–V4 hypervariable region of the bacterial 16S rRNA gene was performed. The amplification of V3-V4 was undertaken with the 341F/805R primer set, and the dual indexing was ensured using unique oligonucleotides within the 2nd amplification. Phusion U Multiplex PCR Master Mix (ThermoFisher Scientific, Waltham, MA, USA) was used for the PCR, followed by magnetic-bead-based purification (Macherey-Nagel, Düren, Germany) after the PCR amplification. Three negative controls were introduced during the PCR amplification and evaluated using agarose gel electrophoresis. The quantity and quality of extracted DNA and amplicon libraries were determined by Qubit Fluorometer (Thermo Fisher Scientific, Waltham, MA, USA) and Agilent 2100 Bioanalyzer systems (Agilent, Santa Clara, CA, USA), respectively. For the sequencing, MiSeq System (Illumina, San Diego, CA, USA) was used with MiSeq Reagent Kit v2 (500-cycles) (Illumina, San Diego, CA, USA) obtaining at least 100,000 paired-end sequencing reads per sample. All samples together with the negative controls were sequenced within the same sequencing batch.

Sequences of V3–V4 primers ca be seen in Table 1.

Water controls in this study are used as blank controls from 2 steps—from extraction of DNA as well as PCR step. Instead of using the sample in DNA extraction control, we added DES solution from FastDNA extraction kit, while in PCR step we used pure water. These controls were included to test purity of work conditions as well as correct true results, as mentioned in section Data Analysis—library preparation contaminants were detected using the “decontam” (v1.18.0).

### 2.5. Data Analysis

Raw sequencing data control was carried out using the “FastQC” (v0.11.9) and “MultiQC” (v1.14) software packages [14]. To increase the proportion of high-quality sequences, sequence filtering based on the adapter content after the V3 and V4 hypervariable region specific “Illumina” primer sequences was performed by excluding said sequences by considering them as sequencing artefacts. At the same time, sequences were trimmed to the length corresponding to average sequence PHRED score quality of at least 20. Both actions were performed with “cutadapt” (v4.14) [15]. “PANDAseq” (v2.11) was used to merge forward and reverse reads, and concatenate unmergeable sequences. Minimums overlap threshold for merging was set at 20 base pairs [16]. Prepared and merged data were denoised using the “DADA2” plugin of “QIIME2” (v2022.2) microbiome bioinformatics platform [17,18]. To further minimize noise, resulting Amplicone Sequence Variants (ASV) were frequency filtered for at least 10 sequences in a sample. Naïve Bayes classifier integrated in the “QIIME2” environment was trained using the “SILVA” (v138.1) rRNA database and used to determine the taxonomic profile of the ASV’s [19]. Library preparation contaminants were detected using the “decontam” (v1.18.0) Bioconductor package was detected by comparing taxa relative frequencies in the real and negative control samples [20]. “phyloseq” (v1.42.0) package was used on the decontaminated data to rarefy to a depth of 3377 sequences to correct for library size differences [21]. Shannon diversity index and Piellou’s index were calculated to evaluate both sample and group-level alpha diversity. Group-level Shannon index values were compared using the Wilcoxon rank sum test. Genus-level PCA was generated to evaluate the inter group differences and similarities using the “microViz” (v0.10.10) package. Finally, differential abundance test between the experimental groups, while correcting for sex as a covariate, was carried out using the “ANCOM-BC2” method from the “ANCOMBC” (v2.0.2) Bioconductor package [22]. All visualizations created using the “ggplot2” (v3.4.2) package.

## 3. Results

Our research included nineteen samples, thirteen male patients and six female patients; with a mean age of 4 years, a standard deviation of 1.47, the oldest patient was 7 years old and the youngest was 3 years old. A Shapiro–Wilk test shows the normality of distribution (W-0.774) (Table 2). The otitis media with effusion group consisted of three male and three female patients. The group with no middle ear effusion consisted of ten male and three female patients. A more detailed demographic distribution can be seen in Table 3.

### 3.1. Alpha and Beta-Diversity

Alpha diversity for both groups was investigated using Piellou’s evenness index and Shannon’s diversity index (Figure 1). A median value comparison was performed using a Wilcoxon signed-rank test, with no significant statistical difference between samples collected from patients with middle ear effusion and samples from patients with no effusion in the tympanic cavity. Beta diversity is represented in Figure 2; a partial overlap of ellipses represents the similarity of the microbiomes of both our study groups signaling a trend towards the studied groups having no statistically significant difference in their bacterial diversity. Statistical analysis using PERMANOVA does not provide strong evidence that the two study groups have different bacterial diversity *p*-0.093.

No statistically significant differentially abundant taxonomic units were detected when comparing both groups.

### 3.2. Taxonomy

The phylum-level taxonomy shows a similar pattern in both groups with *Proteobacteria* and *Bacteroidata* dominating other bacterial phylum (Figure 3). We do observe a significant amount of *Fusobacteriota* and *Firmicutes* in the adenoid surface samples taken from the patients with no middle ear effusion. The samples of adenoid surfaces of patients with middle ear effusion have a much more prolific representation of *Bacteroidota.* One of our samples for the group of patients with no middle ear effusion was dominated by *Actinobactriota* phylum; this phylum was not widely represented in any other group and can be considered an anomaly or contamination by adjacent (dental/buccal) flora. The same can be noted about one sample containing a significant amount of *Sprirochaetota* genetic material, which is not present in such abundance in other samples.

The genus-level taxonomy divided by groups is detailed in Figure 4. The most identified bacterial genera on the adenoid surface in the group with no middle ear pathologies were *Fusobacterium* (24% of total genetic material), *Haemophilus* (22%), *Moraxella* (10%), and *Streptococcus* (11%), also notable bacterial genus were *Prevotella*, *Actinomyces* and *Ralstonia* (all less than 10% of all genetic material).

The bacterial genera from the group that contained children with otitis media with effusion shows a dominance of *Haemophilus* (24%), *Streptococcus* (15%) and *Bordtella* (16% of all genetic material). Also represented in a significant capacity is *Ralstonia* (6%) and *Prevotella* (5%).

## 4. Discussion

We acknowledge that our study groups are uneven with a limited number of samples in each group, which may cause statistical insignificance upon performing statistical tests. Unfortunately, 16S rRNA studies are significantly more abundant and time consuming than classic agar-based microbiological identification methods. We had to deal with budget constraints as well as time constraints knowing the time required to perform proper genetic sequencing [23,24,25]. We consider our findings to show a trend towards adenoid surfaces’ bacterial compositions being more similar than radically different between patients with middle ear effusion and patients with no effusion in the tympanic cavity, despite the lack of statical strength. Obviously, a larger sample pool and a more even group composition may provide the statistical strength to make a stronger statement regarding this issue. 

We used three negative samples to test our DNA extraction and amplification, but we do lack a dedicated control sample pool for the microbiome itself, preferably from healthy individuals. We must, however, point out that acquiring uncontaminated nasopharyngeal swabs of children without symptoms of upper respiratory tract infections is problematic due to the inefficiency of topical anesthesia in the nasal cavity and almost certain contamination by nasal microflora during the swab tip extraction. Uncontaminated nasopharyngeal swabs through the mouth are also, practically, impossible to acquire with no general anesthesia involved.

The lack of dental microflora in our samples shows a significant improvement in our oral disinfection technique in comparison with our previously published results where contamination by dental and gingival microflora, namely *Fusobacterium* and *Shaalia*, was prolific [26]. As seen in our genus level taxonomy, no specific dental microflora was present. This also proves that there is no need for endoscopic endonasal swabs of the adenoid surface to avoid contamination, a thorough oral disinfection is enough.

Our alpha-diversity indexes (Piellou and Shannon) showed a trend towards a lack of significant differences in the composition of bacterial colonies among our two groups; other studies covering this topic show a slight difference in the beta diversity of bacterial colonies on the adenoids among different patient groups [27]. Differences in the Shannon’s index were reported in a study like ours. The authors reported a difference in the Shannon’s index between one study group containing adenoid swabs of children without middle ear effusion and adenoid swabs of children with otitis media. The Shannon’s index was lower in the middle ear effusion group indicating a lower bacterial diversity in the nasopharynx [23].

Comparing alpha diversity between obstructive sleep apnea patients and the otitis media with effusion patient group authors, report a statistically insignificant difference between the mentioned groups with sleep apnea patients displaying a slightly higher alpha diversity [28].

Alpha diversity studies also indicate an association between adenoid microbiome and different clinical statuses and medication effects. For example, authors have suggested a reduction in alpha diversity with nasal steroid use and changes in diversity with age [29]. Our findings, unfortunately, account for different topical medication use and we cannot corroborate these findings. We must, however, note that selecting pediatric patients, who must receive adenoid operations, with no recent history of nasal steroid medication usage can be a challenge due to nasal steroids acting as a primary treatment for several adenoid hyperplasia-associated conditions.

The most identified phylum was Proteobacteria, which coincides with the findings of other authors [23,30]. Proteobacteria phylum itself has been recently renamed *Pseudomonadota* [31], though we use the old term. 

Despite the shift to bacterial genome-based studies, classic culture-based adenoid microbiome studies are also extremely valuable because they allow us to acquire data on antibiotic resistance. The results of these studies also correspond with our studies regarding the composition of bacterial species on the surface of adenoids, with the most common phyla being *Streptococcus* spp., *Haemophilus*, and *Moraxella* [32,33]. The resistance to penicillin-based antibiotics of the *Steptococcal* species was approximated to be a 13%-resistant specimen.

*Streptococcal* carriage is also linked to the dominance of *Alloprevotella*, *Staphylococcus*, *Moraxella*, and *Neisseriaceae* [28]. Our samples did not contain a sufficient amount of either *Neisseriacae* or *Staphylococcus* but we did see a significant amount of *Moraxella* and a reportable amount of *Allopervotella* in our samples. 

*Fusobacterium* is a bacterium commonly identified in children suffering from obstructive sleep apnea [34]. *Fusobacterium* is also associated with anaerobic infections of the upper airways [35]. Our findings place *Fusobacterium* mostly in the group of samples collected from children that had no middle ear effusion. These were mostly sleep apnea children; our findings may indicate the role of *Fusobacterium* in sleep-associated disorders to a bigger degree than middle-ear disorders. 

The dominant genera in our samples were *Haemophilus* and *Moraxella.* This finding can be explained by the distribution and acceptance of the pneumococcal vaccine. Several authors attribute the shift in otitis media etiological pathogens to *Haemophilus influensae* and its domination among other bacterial species, *Moraxella cattarhalis* being the other benefactor of the PCV vaccine distribution [36,37,38]. Our findings suggest the rise of *Haemophilus* and *Moraxella* abundance in comparison with *Streptocuccus* and the effectiveness of PCV vaccines in supering the growth of Streptococcal colonies in the nasopharynx. 

*Hamophilus* dominance in the adenoids was also linked to pediatric obstructive sleep apnea. Children suffering from snoring who also had adenoidal hypertrophy identified *Haemophilus* abundance to a larger degree in comparison with children with sleep apnea but without adnoid hypertrophy [39]. *Haemophilus* is also closely associated with bronchial asthma; the carriage of this bacteria may promote chronic wheezes and shortness of breath in children with recurrent upper airway infections and pneumonia [40,41]. 

Large amounts of *Prevotella* genetic material in conjunction with *Veillonella* and *Streptococcus* on the adenoid surfaces have been attributed to obstructive sleep apnea in children [42]. Our results display *Streptococcus* in relative abundance in both groups and a modest but reportable amount of *Prevotella* genetic material, but *Veilonella* is almost not identified in our samples. 

Regarding the pathogenesis of otitis media with effusion, *Helicobacter pylori* colonization due to gastric reflux is considered to play a critical role in the chronic inflammatory processes of the tympanic cavity. Several studies identify *H. pylori* in the middle ear fluid [43,44]; other studies show no evidence of this bacteria either in the middle ear fluid or in the nasopharynx and express doubt in its significance [45,46]. We found no evidence of *Helicobacter* species in either of our study groups. We may attribute this to the lack of pharyngeal reflux symptoms in our patients. We may speculate that the colonization of the nasopharynx by *H. pylori* is common in patients with pharyngolaryngeal reflux, but more data are required to connect it to the development of otitis media with effusion.

The application of 16S rRNA genetic sequencing to identify bacteria on various organic surfaces is becoming widespread in recent years; other methods of metagenomic analysis is also used for these purposes. One of these methods is “shotgun” metagemomic analysis, which focuses on sequencing and analyzing all the genetic material present in a sample, without specifically targeting any gene or region. It provides a comprehensive view of the microbial community’s genetic content, including the genomes of bacteria, archaea, viruses, and other organisms present in the sample. This analysis may provide us with information on both taxonomy and antibiotic resistance. Unfortunately, “shotgun” metagenomenomic analysis is considerably more expensive than the 16S rRNA identification method we used, thus limiting our resources and reducing our sample to an even smaller size.

## 5. Conclusions

We observed the trend towards bacterial composition on the surface of the adenoids in children with otitis media with effusion and children with no middle ear pathologies to be similar, rather than radically different, despite the lack of statistical power.

The bacterial colonies of both groups consist mostly *of Haemophilus*, *Sptreptococcus*, *Fusobacterium*, and *Moraxella*, with no meaningful differences in bacterial composition between both groups. Our findings may suggest a lesser role of the adenoid microbiome in the pathogenesis of otitis media with effusion.

We acknowledge that our study groups are uneven; a bigger pool of samples may lead to a different outcome. Other studies with similar, albeit slightly larger, sample sizes show more significant differences in identified bacteria among studies groups with different alpha and beta diversity. 

## Figures and Tables

**Figure 1 microorganisms-11-01955-f001:**
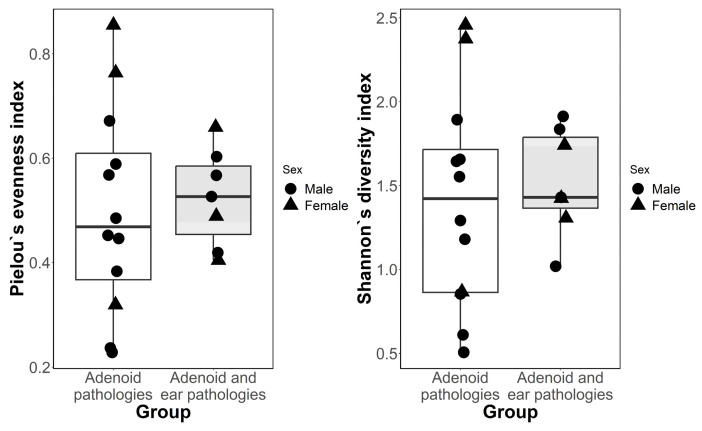
Alpha diversity, Pielou’s evenness index and Shannon’s diversity index. Boxplots shows median values and interquartile values of both study groups. No statistical difference identified between both groups when performing the Wilcoxon rank sum test (*p*-adjusted > 0.05).

**Figure 2 microorganisms-11-01955-f002:**
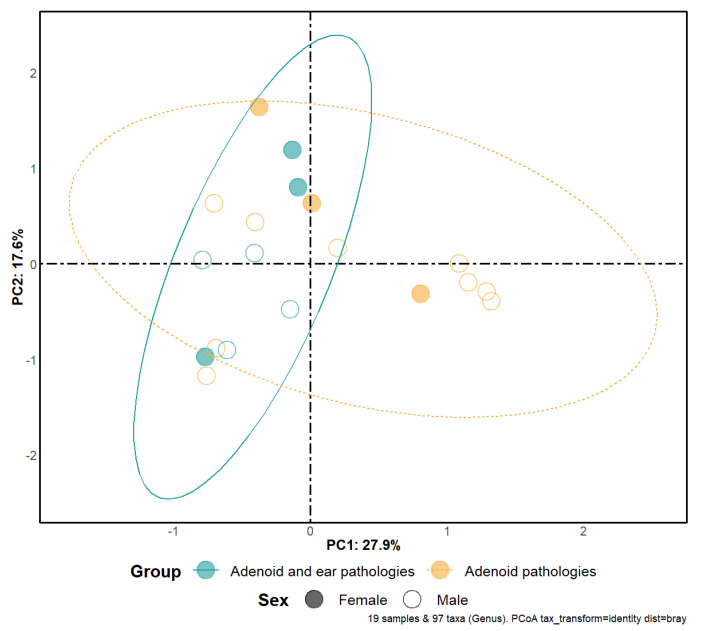
Beta diversity, plot shows a comparison between two study groups. Different colors represent different subgroups (male and female). Separation of samples is based on principal components and confidence level is set to 95%. We observe a partial overlap of ellipses signaling a trend towards similarities between the studied groups.

**Figure 3 microorganisms-11-01955-f003:**
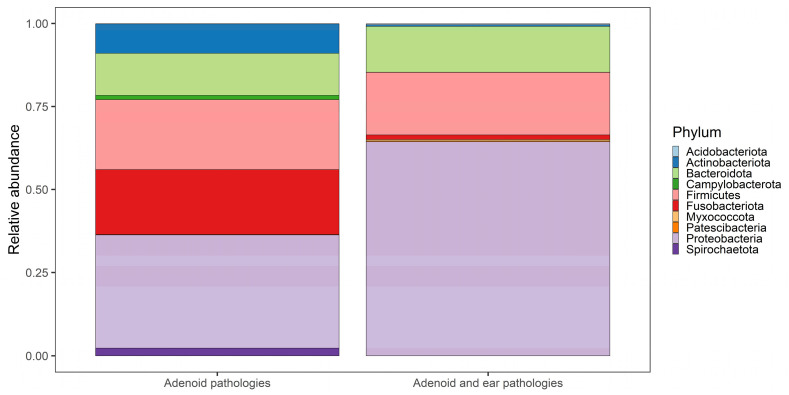
Taxonomy bar plot representing sample level relative abundances of their taxonomic composition at phylum level. Samples sorted according to their comparison group. Taxa comprising the rest of the relative composition not included in the plot.

**Figure 4 microorganisms-11-01955-f004:**
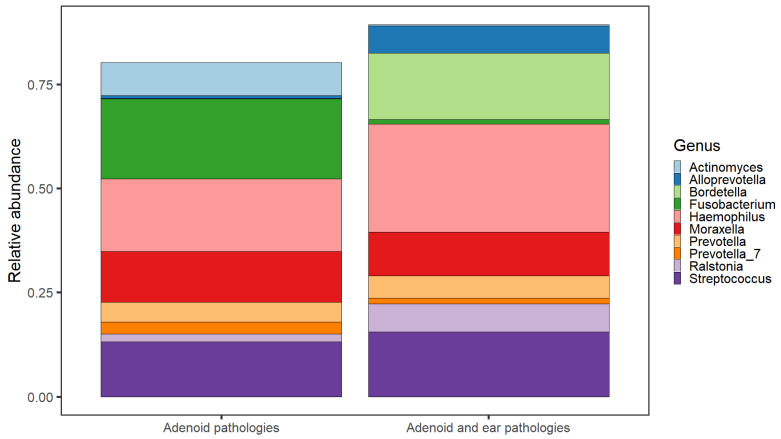
Taxonomy bar plot representing the relative abundances of the ten most prevalent taxa in the respective comparison groups of ‘Adenoid pathologies’ and ‘Adenoid and ear pathologies’. Taxa comprising the rest of the relative composition not included in the plot.

**Table 1 microorganisms-11-01955-t001:** Sequences of V3-V4 Primers.

Name	Sequence
ci5_16S_V3_Fw(341F)	TCGTCGGCAGCGTCAGATGTGTATAAGAGACAGNNNNNNCCTACGGGNGGCWGCAG
ci7_16S_V4_Rs(805R)	GTCTCGTGGGCTCGGAGATGTGTATAAGAGACAGNNNNNNGACTACHVGGGTATCTAATCC

**Table 2 microorganisms-11-01955-t002:** Descriptive statistics.

Descriptives									
								Shapiro-Wilk	
	N	Missing	Mean	Median	SD	Minimum	Maximum	W	*p*
Age	19		4.20	4.00	1.47	3	7	0.774	<0.001

**Table 3 microorganisms-11-01955-t003:** Demographics table detailing the gender/age distribution among study participants.

	Children with no Middle Ear Effusion	Children with Middle Ear Effusion
Gender	Male—10 participants Female—2 participants	Male—4 participants Female—3 participants
Age	3 y.o.—3 participants 4 y.o.—5 participants 5. y.o.—3 participants 6 y.o.—1 participant	3 y.o.—1 participant 4 y.o.—3 participants 5. y.o.—2 participants 6 y.o.—1 participant

## Data Availability

All data are available upon reasonable request from the corresponding author.

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
