# Peer review of "Identifying the Microbiome of the Adenoid Surface of Children Suffering from Otitis Media with Effusion and Children without Middle Ear Effusion Using 16S rRNA Genetic Sequencing"

_microorganisms, 2023, doi:10.3390/microorganisms11081955_

Round 1

Reviewer 1 Report

Identifying the microbiome of the adenoid surface of children suffering from otitis media with effusion and children with healthy ears using 16s rRNA genetic sequencing

Summary:

The study describes the analysis of microbial diversity, composition, and abundance among children patients with otitis media with effusion compared to controls using 16S ribosomal RNA sequencing from the adenoid surface. 6 cases and 13 controls are included in this study. No significant differences are identified.

Major comments:

-       Despite this issue being briefly discussed later on in the manuscript, the fact that controls might be more susceptible to otitis media with effusion – they had indications for adenoid surgery – should be emphasized. Indeed, as susceptible patients, one would expect their microbiota to be closer to those with otitis media with effusion. Using the term “healthy ear” might additionally be misleading. I would recommend referring to patients healthy, but at risk, rather than just healthy.

-       The fact that no significant differences were detected is very likely to stem from the low number of cases included. Even though this is reported briefly in the text, I would advise the authors to rephrase the corresponding sections. Indeed, the authors should report an estimate of power, or, if not feasible, rephrase the finding reports so that trends were identified, but were not significant, and that this might be due to power issues. Currently, the text reads as if there are no differences, but that cannot be tested with these numbers, and thus, should be completely reworked.

-       Water controls are mentioned, but it is unclear how these are used in the pipeline, and if any correction of the patient sample microbiota is carried out using those.

-       It is key that a much more detailed demographics table is added. A table split by group, and, including, at least, sex, disease onset (if available), antibiotic use, co-morbidities, and indication for surgery in control patients, needs to be included.

-       Regarding the beta diversity figure, usually when observing such a pattern, one would argue that there seems to be a difference. I would advise the authors to consider rephrasing. There is certainly an overlap, but it is only partial.

-       The results of differential abundance analyses, using ANCOMBC, should be reported as a figure, or as a supplementary table, perhaps. It is important that this is shown and discussed in more detail.

-       A discussion on how metagenomics could be used in this setting would provide a more complete picture. At the moment, only culture-based and 16S-based technologies are mentioned.

Minor comments:

-       A list of prevalent organisms is reported in the abstract. I believe this refers to the whole population, rather than just one of the groups. This should be made more explicit.

-       A reference for reporting that alteration of the microbiota can lead to otitis, in the introduction, should be added to support the claim.

-       In the results, it seems that there is typo when reporting the standard deviation for age. 14.7 should read 1.47.

-       In Figure 1, the colors should be uniform. Indeed, in one of the panels, adenoid pathologies are reported as purple-colored dots and as red-colored dots in the other panel.

-       The comment on distribution disproportion should be removed as it is completely speculative and uninformative here.

-       When reporting the PERMANOVA p-value, there is a typo in 0.09.3.

-       The legend for Figure 1 should be directly under Figure 1.

-       Unless I misunderstood, it appears the authors mention a PERMANOVA p-value for diversity. This is quite uncommon and a justification would be helpful.

-       There are two paragraphs labeled 3.1.

-       In Figure 3, there is a Prevotella label and a Prevotella_7 label. Could the authors explain how those are different?

-       For consistency, Figures 3 and 4 should follow the same style, perhaps summarized by group, with detailed per-sample results as supplementary.

-       On figure 3, the abundance bars do not go all the way up to 1. Could the authors explain why?

-       In the discussion, was “expansive” meant as “expensive”?

/

Author Response

To the esteemed reviewer

We are extremely grateful to you for dedicating your time and providing insightful feedback on our manuscript.Here are ways we improved our manuscript based on your valuable suggestions:

  • Despite this issue being briefly discussed later on in the manuscript, the fact that controls might be more susceptible to otitis media with effusion – they had indications for adenoid surgery – should be emphasized. Indeed, as susceptible patients, one would expect their microbiota to be closer to those with otitis media with effusion. Using the term “healthy ear” might additionally be misleading. I would recommend referring to patients healthy, but at risk, rather than just healthy

We absolutely agree that the term “healthy ear” is misleading and adjusted both our title to exclude “healthy ear” and changed it to “children without middle ear effusion” and changed the “healthy ear” to “children with no middle ear effusion” We have also included a text regarding adenoid surgeries for children with no middle ear pathologies that emphasizes the significance of removing adenoid tissue to prevent middle ear effusion forming. 

  • The fact that no significant differences were detected is very likely to stem from the low number of cases included. Even though this is reported briefly in the text, I would advise the authors to rephrase the corresponding sections. Indeed, the authors should report an estimate of power, or, if not feasible, rephrase the finding reports so that trends were identified, but were not significant, and that this might be due to power issues. Currently, the text reads as if there are no differences, but that cannot be tested with these numbers, and thus, should be completely reworked

We also acknowledge the limited sample size, unevenness between the groups and lack of our statistical power, unfortunately, due to monetary and time constraints this is the best we could do in our circumstances. We have adjusted the text as you suggested, in the following ways:

  • We adjusted the abstract in the results section emphasizing a trend rather than hard statistical evidence.
  • We re-wrote the discussion section also emphasizing a trend towards statistical similarity rather than a hard statement and suggests a larger sample pool in the future

  • Water controls are mentioned, but it is unclear how these are used in the pipeline, and if any correction of the patient sample microbiota is carried out using those

Regarding the water controls in this study are used as blank controls from 2 steps – from extraction of DNA as well as PCR step. Instead of sample in DNA extraction control, we added DES solution from FastDNA extraction kit, while in PCR step we used pure water. These controls were included to test purity of work conditions as well as correct true results as mentioned in section Data analysis - Library preparation contaminants were detected using the “decontam” (v1.18.0).

We added this description in the text of the manuscript in the DNA extraction section, line 212

  • It is key that a much more detailed demographics table is added. A table split by group, and, including, at least, sex, disease onset (if available), antibiotic use, co-morbidities, and indication for surgery in control patients, needs to be included

Regarding a more detailed demographic table, we believe an additional table or a bigger demographic table that we have would not improve our manuscript due to low sample size. The gender distribution is mentioned in text, and, in our opinion, gender cannot influence bacterial composition, the age is mentioned in the table we provided. Since our middle ear effusion group is 6 samples, making a table for it seems excessive. However, we added the gender distribution inside the groups in the text before the table as you suggested.

As we mentioned in our inclusion and exclusion criteria, we do not admit patients with severe/note-worthy co-morbidities as well as patients receiving antibiotic treatment that may influence microflora are not included in our study. A detailed patient history is also not reliable as it is subjective. Indications for surgery for both groups are also detailed in the materials and methods section.

  • Regarding the beta diversity figure, usually when observing such a pattern, one would argue that there seems to be a difference. I would advise the authors to consider rephrasing. There is certainly an overlap, but it is only partial

We adjusted the beta diversity description emphasizing the partial overlap not complete overlap and including a mention of a trend towards no statistical difference rather than a hard statement. We re-iterate this in the text before the figures.

  • The results of differential abundance analyses, using ANCOMBC, should be reported as a figure, or as a supplementary table, perhaps. It is important that this is shown and discussed in more detail

Commenting on the differential abundance analyses, we described our method in the text and while trying out different visual representation found the figures produced uninformative and hard to comprehend, that is why we did not include and extra figure or a supplementary table.

  • A discussion on how metagenomics could be used in this setting would provide a more complete picture. At the moment, only culture-based and 16S-based technologies are mentioned

We consider 16s rRNA genetic testing as a part of metagenomic analysis, with 16S rRNA sequencing is a specific approach within the larger field of metagenomics. Considering you suggestion we elaborated another metagenomic method of identifying bacterial DNA such as shotgun method in our discussion section in the end, line 403

  • A list of prevalent organisms is reported in the abstract. I believe this refers to the whole population, rather than just one of the groups. This should be made more explicit

Corrected the part in the abstract to signal that the mentioned bacteria are referred to both groups and elaborated, line 32.

  •  A reference for reporting that alteration of the microbiota can lead to otitis, in the introduction, should be added to support the claim

Reference added in the introduction to support the claim as you asked.

  • In the results, it seems that there is typo when reporting the standard deviation for age. 14.7 should read 1.47.

Corrected the deviation age to 1.47, thank you for pointing it out.

  • -In Figure 1, the colors should be uniform. Indeed, in one of the panels, adenoid pathologies are reported as purple-colored dots and as red-colored dots in the other panel

Regarding the coloring of the figures, our tools for generating tax plots and diversity ellipses suggest this coloring as the most used and practical, which is why we are using this coloring scheme.

  • The comment on distribution disproportion should be removed as it is completely speculative and uninformative here

Removed the distribution disproportion as suggested.

  • When reporting the PERMANOVA p-value, there is a typo in 0.09.3.

Corrected the PERMANOVA typo to 0.093.

  • The legend for Figure 1 should be directly under Figure 1

Corrected the Figure 1 picture so that it is directly over the legend aligning with the text.

  • Unless I misunderstood, it appears the authors mention a PERMANOVA p-value for diversity. This is quite uncommon and a justification would be helpful

Regarding the justification for using PERMANOVA, it is of nonparametric nature, a method that does not make assumptions about the distribution of the data as our data is not distributed evenly. This makes it suitable for analyzing microbial community data, which can often exhibit complex and non-normal distributions.

  • There are two paragraphs labeled 3.1”

Corrected the double 3.1 chapter renaming the second one 3.2 Diversity

  • In Figure 3, there is a Prevotella label and a Prevotella_7 label. Could the authors explain how those are different?

Commenting on Pervotella_7 - These bacteria SILVA database identifies as different genus based on differences between their sequences.

  • For consistency, Figures 3 and 4 should follow the same style, perhaps summarized by group, with detailed per-sample results as supplementary

Figure 3. and 4. follow the same visual style, however, they represent different taxonomic resolution at different taxonomic levels. While Figure 3. represents 10 most abundant genus level taxa per group, Figure 4. represents microbial composition at the Phylum level for each of the samples accordingly giving more resolution.

  • On figure 3, the abundance bars do not go all the way up to 1. Could the authors explain why?

The relative abundance bars don`t go all the way up to 1 because this figure displays the 10 most abundant genera for the respective groups, accordingly, they don`t necessarily make up the whole microbiome composition at the genus level for these samples and groups, therefore they don`t have to and in this case they don`t reach the relative abundance value of 1.0 or 100%

  • In the discussion, was “expansive” meant as “expensive”?

Expansive as a derivative of the word expanse – meaning wide or abundant, but we changed the “expansive” term to abundant.

We hope our corrections in the manuscript are suitable enough to allow our work to be published, once again thank you for your valuable insight!

The authors of the manuscript

Reviewer 2 Report

This manuscript aimed to assess the assess the most common bacteria present on the surface of adenoids in children with otitis media with effusion and compare them to children without pathologies in the tympanic cavity. This manuscript addresses important point in the field that aim to identify the source of respiratory microbiota. However, some comments need to be addressed.

Major comment. The sample size is not enough to address the address in microbiome composition. This study also doesn’t include positive control samples to assess the contamination.  However, the authors acknowledge this as a limitation for this study which is a plus.

Have the authors assessed the core microbiota between the two groups?

Minor comments.

Line 17. Replace the word accumulation with other suitable word.

Line 26 The most prevalent phyla in which group?

Line 27. The most common bacterial genera in which group?

Line 28. Name of genera should be italic

Line 30. Microorganism diversity. Is this alpha or beta diversity.

The introduction section needs to be improved. There is no connection between paragraphs.

 Line 155. Use samples instead of materials.

Line 175. Please include the sequence information for the V3-V4 primers.

Line 179. What type of negative control samples were used?

Line 221. Piellou’s evenness index have not mentioned in the method section.

Line 256. Genera name should be italic.

I think acknowledging these comments, I believes the manuscript would be suitable for publications after minor revision.

Author Response

To the esteemed reviewer,

We sincerely appreciate the time and effort you have dedicated to reviewing our manuscript. In this letter, we address each of your comments and suggestions, providing explanations and outlining the revisions we have made.

  • Major comment. The sample size is not enough to address the address in microbiome composition. This study also doesn’t include positive control samples to assess the contamination. However, the authors acknowledge this as a limitation for this study which is a plus

We absolutely agree that our sample size is, relatively, small but in the range of other nasopharyngeal microbiome based 16s rRNA published studies:

For example: https://pubmed.ncbi.nlm.nih.gov/31218867/

Unfortunately, due to budgetary constraints we are currently unable to increase our sample size.

Regarding the positive control samples – unfortunately due to monetary and time limitations we are unable to expand our sample pool to include different types of control samples, we used water for negative control samples.

  • Have the authors assessed the core microbiota between the two groups?

The swabs were taken from the surface of the adenoids rather than the core as we believe it adequately represents the bacterial composition and additional damaging of the tissue by cutting it may disrupt the microbial colonies.

  • Replace the word accumulation with other suitable word.

Changed the word accumulation to the word colonization as per your suggestion (line 19)

  • The most prevalent phyla in which group?

We adjusted most common phyla in the results section.

  • The most common bacterial genera in which group?

Most common bacterial genera in each group are mentioned in the text, we believe an extra figure is unnecessary as it is just an iteration on the existing figure.

  • Name of genera should be italic

Adjusted the font in genera naming to italics in the text.

  • Microorganism diversity. Is this alpha or beta diversity

Detailed in the text where diversity means beta diversity, instead of ambiguous diversity.

  • The introduction section needs to be improved. There is no connection between paragraphs

Re-wrote the introduction section so that it is more cohesive.

  • Use samples instead of materials

Changed material to samples in line 178

  • Please include the sequence information for the V3-V4 primers

We included sequence information for the V3-V4 primers in the text in the methods section, line 247

  • What type of negative control samples were used?

Regarding the negative control sample usage – we used water samples as negative control to test our method, we detailed it in the DNA extraction section

  • Piellou’s evenness index have not mentioned in the method section

Added Piellou’s index in the data analysis section line 239

  • Genera name should be italic

Genera names were corrected in italics

Once again, we sincerely appreciate the time and effort you have devoted to reviewing our manuscript. Hopefully, the changes we made in our manuscript based on your insight will improve our work

The authors of the manuscript

Round 2

Reviewer 1 Report

I thank the authors for their thorough comments and edits. I believe the manuscript is much improved and that this interesting study will only be more impactful. I have a few further comments that I think would only improve the manuscript and that the authors might consider incorporating.

Major comments:

-       I really believe, even considering the small number of samples, that a demographics table would be a valuable addition. As a reader, from a clinical perspective, I consider it very helpful if I can have a quick overview of the cohort and estimate how diverse the population is. This usually provides compelling evidence as to how reproducible the findings would be in a different setting and/or population.

-       I understand that including all the results from ANCOMBC might provide a noisy picture however, for transparency and completeness, I believe those should be reported in some form. A volcano-like plot could be used or perhaps, a supplementary table could be added. 

Minor comments:

-       My apologies if I have missed the new reference for reporting that alteration of the microbiota can lead to otitis in the introduction, but I would expect it to be reported on lines 82-84.

-       In Figure 1, I find the color scheme to be appropriate and have no comments regarding the color choice. However, each group should have the same color associated to it across the manuscript i.e., adenoid pathologies should have the same color across figures, and this is currently not the case. It is purple in some of the figures and red in some of the others, and this can be quite confusing for the reader.

-       In Figures 3 and 4, I understand that different taxonomic levels are represented. This provides valuable information however, why not use the same grouping rationale i.e., summarized by group for both or by sample for both? These would be better for interpretation from a reader’s perspective.

Author Response

To the esteemed reviewer,

We would, once again, thank you for giving valuable insight on how to improve our manuscript. We carefully analyzed your suggestions and here are ways we improved our work:

  • An additional table has been added (Table 3) with all the demographic information we currently have. We understand it could be more detailed, however we would like to stress that we do not consider several factors you mentioned in your review:
  • Children receiving antibiotic treatment for the last 2 wreaks are not included in the study (see exclusion criteria)
  • Children with note-worthy/ severe co-morbidities are also not included, since we are an out-patient clinic
  • Vaccination status is unreliable in our setting as Latvia currently does not have a centralized vaccination status database and all information on vaccination status cannot be verified unless we specifically make blood-tests to evaluate antibodies

  • Regarding ANACOMBS results and addition box-plots or figures – our results detect no statistically significant differentially abundant taxonomic units when comparing both groups, we included this phrase in line 279-280.

We believe that a detailed ANACOMBC figure is necessary only when statistically significant difference is achieved to identify the distinguishable factor, in our case no statistically significant taxonomic unit was identified thus an extra figure which conveys no note-worthy information will only make the manuscript more cluttered.

  • We are sorry if our text is a little bit misleading, you are probably referring to the phrase in line 65-66 where we state that changes in oral microbiota can increase the risk of the development of several aural and nasal conditions. We do imply that alterations in the microbiome lead to middle ear conditions as a hard fact, even our conclusions state otherwise. Nevertheless, we added a reference (7) in line 68
  • We adjusted Figure 1 to make it black and white to make colors match among Shannons and Piellous indexes. We are sorry, but we dot completely understand your suggestion regarding the choice of color among all figures.

In the articles we analyzed color schemes among the figures were different in each figure, for example :

https://www.ncbi.nlm.nih.gov/pmc/articles/PMC6937193/

https://www.ncbi.nlm.nih.gov/pmc/articles/PMC6937193/

https://journals.sagepub.com/doi/full/10.1177/01455613221135647?rfr_dat=cr_pub++0pubmed&url_ver=Z39.88-2003&rfr_id=ori%3Arid%3Acrossref.org

  • We changed the Figure 4 as you suggested

Hopefully, these adjustments will improve our manuscript to make it suitable for publishing.

Best wishes,

The authors of the manuscript